# Reconstruction of Cylindrical Surfaces Using Digital Image Correlation

**DOI:** 10.3390/s18124183

**Published:** 2018-11-29

**Authors:** Adilson Berveglieri, Antonio M. G. Tommaselli

**Affiliations:** 1Department of Statistics, São Paulo State University UNESP, 305, Presidente Prudente 19060-900, Brazil; 2Department of Cartography, São Paulo State University UNESP, 305, Presidente Prudente 19060-900, Brazil; a.tommaselli@unesp.br

**Keywords:** cylinder, geometric transformation, LSM, optical measurement, reconstruction

## Abstract

A technique for the reconstruction of cylindrical surfaces using optical images with an extension of least squares matching is presented. This technique is based on stereo-image acquisition of a cylindrical object, and it involves displacing the camera following the object length. The basic concept behind this technique is that variations in the camera viewpoint over a cylindrical object produce perspective effects similar to a conic section in an image sequence. Such parallax changes are continuous and can be modelled by a second-order function, which is combined with an adaptive least squares matching (ALSM) for the 3D object reconstruction. Using this concept, a photogrammetric intersection with only two image patches can be used to model a cylindrical object with high accuracy. Experiments were conducted with a cylinder on a panel with coded targets to assess the 3D reconstruction accuracy. The accuracy assessment was based on a comparison between the estimated diameter and the diameter directly measured over the cylinder. The difference between the diameters indicated an accuracy of 1/10 mm, and the cylindrical surface was entirely reconstructed.

## 1. Introduction

A dense 3D reconstruction can be performed using active sensors or image data coming from passive sensors. Laser scanners or structured light systems are commonly used to generate dense point clouds, as these systems can collect large volumes of points in a short time even over textureless surfaces. Other techniques to obtain dense point clouds use images that are collected by passive sensors. Advances in sensor technology and the development of new dense matching algorithms have allowed the generation of dense clouds of points [1,2,3] with passive optical images, which provide high precision, reliability and automation. Significant improvements in the hardware and algorithms, e.g., Structure from Motion (SfM), have made photogrammetry a competitive technology due to its high accuracy, automation and affordable costs. Objects can be successfully imaged and reconstructed using image-based approaches, and 3D modelling can achieve results that are comparable or even better than those achieved from laser scanning.

Image matching is a fundamental task in the dense object reconstruction procedure, and it requires the establishment of correspondences between primitives that are extracted from images. The problem of image matching can be solved by using stereopsis [4,5] or multi-image correspondences [1,6,7,8]. In recent decades, the strategies for dense point correspondences have been focused on multi-view approaches [9,10], graphics processing units (GPUs) [11], per-pixel measurements [12], global energy minimisation algorithms [5,13], and dynamic programming approaches [14]. A review on high density image matching was presented by Remondino et al. [15] in which available open-source and commercial software were also assessed for dense point cloud generation.

Several matching techniques, such as feature- and area-based algorithms, have been proposed in the photogrammetric literature. Among them, least squares matching (LSM) is a highly accurate area-based matching technique that is well-known and is used to set up a geometric correspondence of two or more image patches. The basic concept was introduced by Förstner [16] and later refined by Ackermann [17] and adapted for image matching by Gruen [18]. Ackermann [17] and Gruen [19] evaluated the use of geometric and radiometric parameters in rectangular and square patches in which an affine transformation was used as a linear model. Furthermore, adaptations of LSM have been tested and assessed in practical usability, e.g., using multi-image adjustment with additional geometric constraints [20] and approaches for object-oriented matching [21]. Other studies have investigated the adaptability and performance of the LSM, as performed by Bethmann and Luhmann [22], who used an approach with a projective transformation in the function model to improve the adaptability.

Regarding the application of LSM for matching refinement, Zhang et al. [23] presented a scheme for matching keypoints in images acquired by unmanned aerial vehicles (UAVs). An LSM based on pyramids was used as a refinement step to improve the final precision. Debella-Gillo and Kääb [24] explored LSM applications related to surface displacement and deformation of mass movements. The authors showed that LSM could match the images and strain rates accurately.

Matching algorithms have been developed along with studies on their accuracy, as reported in the first experiments produced by Schewe and Förstner [25] in industrial applications and later in research using DSM generation [26,27,28,29,30]. Xu et al. [31] conducted studies on the measurement accuracy and efficiency using compensation methods based on digital image correlation (DIC). Other studies have also been conducted to evaluate the accuracy achieved with image-based methods, for example, for deformation measurements [32,33], shape measurement [34] and deflection in cylindrical structure [35]. Remondino et al. [15] commented on the difficulties in defining an evaluation element (entire surface, small patches or points) and which procedure to use for accuracy assessment. For example, Seitz et al. [36] assessed the completeness and accuracy based on the Euclidean distance (which indicates how much of the scene was reconstructed and how close to the ground truth the result was). Bethmann and Luhmann [22] indicated some problems with the performance of LSM, which could affect its results, such as the texture, template size, geometric distortion between images, quality of the initial values and transformation model for matching. In contrast, improvements achieved with technological advances have sped up photogrammetric tasks and produced results that have better quality and accuracy at an affordable cost, which makes photogrammetry an attractive field for research and widespread application. Sutton et al. [37] presented a discussion considering recent progress in DIC.

This paper introduces a novel technique to produce a continuous 3D reconstruction of a cylindrical surface based on stereo-pairs. When an image sequence over a cylindrical object is collected by moving a camera in a single direction, image patches of corresponding areas are distorted according to a plane projection of a conic section. These distortions make it unfeasible to perform image matching for large image patches. The problem to be solved is the correction of geometric distortions to allow the application of a continuous matching, thus making it feasible to reconstruct the cylindrical surface. The hypothesis was that it would be possible to model the parallax changes by a geometric transformation function to correct the distortion in such a way that homologous areas could be matched and refined with a modified LSM to achieve sub-pixel accuracy. Consequently, all pixels of a continuous patch of the cylindrical surface could have their 3D coordinates estimated.

Moreover, the current techniques typically require a large set of images to perform a 3D reconstruction. In contrast, this proposed technique can use only a single stereo-pair (two images) to generate a 3D point cloud of a cylindrical surface with high accuracy. The main objective of this study is to present and assess the reconstruction technique using a modified function for the LSM in cylindrical surfaces. The experiments were based on a cylindrical object in the laboratory for conceptual valuation purposes, but the approach can be extended to applications with similar surfaces such as construction pipes, lighting poles, mechanic parts, and other objects. The next sections present the methodological steps, experiments and results assessment, showing that an accuracy of 1/10 mm can be achieved in similar datasets.

## 2. Methodology

The proposed methodology aims at performing a continuous reconstruction of 3D cylindrical patches with high accuracy. The technique accomplishes a fitting between the image patches extracted from the images collected at different viewpoints using a modified geometric transformation for the LSM.

### 2.1. The Concept

Figure 1 depicts the image acquisition procedure to understand the concept. The image acquisition is performed by displacing the camera in a line path parallel to the cylindrical axis. In this example, three views of a strip over the cylinder are presented. Due to the cylindrical shape and the viewpoint changes, three geometric shapes that correspond to the strip over the cylinder are produced in the images. As seen in Figure 1, the strip appears as a horizontal shape (the main axis appears as a straight line) when the image is collected from a normal view (the projecting ray to the strip over the cylinder). If the projecting ray of the strip path is oblique, then the strip has a curved shape.

The camera displacement path parallel to the cylindrical axis assures that the images have approximately the same scale, and differences in the image patches of the object are caused by the depth and viewpoint changes. In this case, to make a 3D reconstruction with an image matching procedure using the frontal image as a reference, it is necessary that one image with a suitable base length has the corresponding pixels geometrically transformed to present similar features between the homologue images, which makes it possible to fit a model, as depicted in Figure 2.

Given two image patches or regions that refer to the same area over the cylindrical diameter but with different perspective views, parallaxes will occur due to depth and orientation variations. A suitable geometric transformation combined with an image matching procedure can relate the corresponding pixels of the two patches to produce a continuous 3D reconstruction of all pixels within the region.

It is important to note that the presented example was based on a vertical camera displacement parallel to the cylindrical axis. However, if the object is lying in a horizontal plane, then the camera movement should be horizontal and the same effect will be produced. The first step is to determine which function can transform a curved patch (a conic section) to a reference patch, with both representing the same area in the object. The reference patch does not need to be exactly perpendicular to the camera because a 3D reconstruction will be made later. Next, a second step is performed to refine the image coordinates to achieve sub-pixel precision, as will be presented in the next two sections.

### 2.2. Mathematical Model

The concept of the proposed technique for cylindrical surface reconstruction uses a geometric transformation *T* with a further refinement by an adaptive least squares matching (ALSM) to accurately map a point from an image *I*_1_(*x*, *y*) to its respective correspondence in an image *I*_2_(*x*, *y*) with sub-pixel precision.

Because the camera displacement is nearly parallel to the cylindrical axis, the mapping *I*_1_(*x*, *y*) → *I*_2_(*x*, *y*) in the x-direction (horizontal) can be made by an affine transformation, which is sufficient to solve small differences because the variations are neglected. In the y-direction, i.e., (vertically), the cylindrical surface produces a perspective effect that is similar to a conic section, which requires a non-linear transformation; this step can be modelled as a parabola. Thus, the mapping in y is made by an affine transformation with the addition of a second-order term, as shown in Equation (1):(1)$$\{\begin{array}{c} x^{\prime }=a_{1}+a_{2}x+a_{3}y\\ y^{\prime }=b_{1}+b_{2}x+b_{3}y+b_{4}x^{2} \end{array}$$where *a*_1_ and *b*_1_ are translations; *a*_2_, *a*_3_, *b*_2_ and *b*_3_ denote parameters that fit the shape (rotation, scales in *x* and *y*, and shear); *b*_4_ represents the concavity of the curve; and *I*′_2_(*x*′, *y*′) are the coordinates transformed from a point *I*_1_(*x*, *y*). If the displacement is performed in the x-direction, a similar model could be used, but introducing an additional parameter in the first line in Equation (1).

The parameters of the geometric function *T* in Equation (1) can be estimated by least squares adjustment [38], provided that a minimum of four suitable corresponding points have been established between the two regions. The pixel coordinates of the image patch of the cylindrical surface in *I*_1_ are used as observations in the system of equations, and the seven parameters (*a*_i_ and *b*_i_) are the unknowns to be estimated. After estimating the parameters, the transformation *T* is applied to the cylindrical region of *I*_1_ to generate coordinates that must be re-sampled over a new grid of discrete coordinates.

This preliminary transformation provides a good approximation between the corresponding regions, minimising the geometric distortions caused by the orientation and depth (curves are reshaped to near-linear forms), which enables the application of the final image matching procedure with ALSM refinement in a following step.

### 2.3. Refinement Using ALSM

Let *f*(*x*, *y*) and *g*(*x*, *y*) denote conjugated image patches of a region over a cylindrical surface after a preliminary geometric transformation. The problem of image matching based on ALSM can be solved by estimating the transformation parameters to match the region *g*(*x*, *y*) with *f*(*x*, *y*). However, due to random effects between the regions, an error vector *e*(*x*, *y*) is added to establish the image matching model, as seen in Equation (2):(2)f(x,y)−e(x,y)=g(x,y)

The position values of *g*(*x*, *y*) must be determined to provide a match point. An approximation to at least a few pixels is required, which typically is accomplished by a correlation coefficient and, in this case, by an initial transformation. Next, the refinement using the transformation *T* in Equation (1) with ALSM is performed based on the technique proposed by Gruen [19]. Because the adjustment of one image region to another is a non-linear problem, the seven parameters are linearized by a Taylor series expansion. The differentiation yields Equation (3):(3)$$\{\begin{array}{c} dx=da_{1}+da_{2}x+da_{3}y\\ dy=db_{1}+db_{2}x+db_{3}y+db_{4}x^{2} \end{array}$$

In addition to these seven parameters, two radiometric parameters, *r*_1_ and *r*_2_ (the brightness and contrast, respectively), are included to form the system of linearized equations in Equation (4) in which *g*′(*x*, *y*) is the re-sampled image region from *g*(*x*, *y*):(4)$$f\left(x,y\right)-e\left(x,y\right)=g\left(x,y\right)=g^{\prime }\left(x,y\right)+\frac{\partial g^{\prime }\left(x,y\right)dx}{dx}+\frac{\partial g^{\prime }\left(x,y\right)dy}{dy}+r_{1}+r_{2}g^{\prime }\left(x,y\right)$$

For simplification, the partial derivatives of *g*(*x*, *y*) in Equation (4), which correspond to image gradients, can be represented by *g_x_* and *g_y_*, as shown in Equation (5):(5)$$g_{x}=\frac{\partial g\left(x,y\right)}{\partial x}\;\mathrm{and}\;g_{y}=\frac{\partial g\left(x,y\right)}{\partial y}$$

The digital number values and coordinates of all corresponding pixels from the regions are used together in Equation (4). Thus, a linearized system with nine parameters in Equation (6) is obtained:(6)$$\begin{array}{c} f\left(x,y\right)-e\left(x,y\right)=g^{\prime }\left(x,y\right)+g_{x}da_{1}+g_{x}xda_{2}+g_{x}yda_{3}+g_{y}db_{1}+g_{y}xdb_{2}\\ +g_{y}ydb_{3}+g_{x}x^{2}db_{4}+r_{1}+r_{2}g^{\prime }\left(x,y\right) \end{array}$$

This system can be solved iteratively by the least squares method using the adjustment proposed by Gruen [19], but we consider in this case an additional parameter, db_4_. After completing the ALSM procedure, all pixels of the conjugated region have their image coordinates matched between the two images with sub-pixel precision. Then, 3D surface reconstruction can be achieved by a photogrammetric intersection procedure with collinearity equations (using intrinsic and extrinsic parameters of cameras). The determination of the 3D coordinates is continually performed pixel by pixel for the cylindrical surface in a local reference system.

In summary, the following steps are used in this procedure:Initial approximation between the conjugated regions with an extraction of distinguishable corresponding points (or keypoints) from the cylindrical surface. These points are used to estimate the seven parameters (*a*_i_ and *b*_i_) of the geometric transformation via least squares adjustment. Some kind of texture is required to extract points that describe the surface curvature, which could be produced either by a speckle painting [33] or by structured light projection.From the matched points, a window is opened around the conjugated region, which covers the cylindrical diameter.The image coordinates of the image patch that has the curve effect is re-sampled by the transformation function that performs the first approximation of the image coordinates with a discrepancy of a few pixels.The correlation coefficient is computed between the image patches to determine a match point at the pixel level, although some distortions are still noticeable.An image matching refinement by iterative ALSM is performed to fine-tune the shape and estimate the sub-pixel coordinates of all points in the region.The continuous 3D reconstruction for all points inside the region is made with a photogrammetric intersection.

The presented technique was implemented and tested. In the following sections, the technical applications and results will be presented, as well as the validation and accuracy assessments achieved with the 3D reconstructions.

## 3. Data and Experiments

The experiments were conducted using images acquired by a digital camera with a conventional lens, and the object to be reconstructed was a cylinder with two different textures.

### 3.1. Camera Calibration

The specifications of the digital camera that was used to collect the images for the experiments are presented in Table 1.

As shown in Figure 3, a panel (90 cm × 120 cm) composed of markers with Aruco codification [39] was used for camera calibration. These markers were automatically detected and recognized using software adapted by Tommaselli et al. [40] to extract the image coordinates of the Aruco corners. The set of image coordinates was used as observations in a bundle adjustment for camera calibration.

A set of 52 images was collected from several camera stations, with different orientations, and used in the camera calibration procedure. Intrinsic camera parameters such as focal length, principal point coordinates, and radial and decentering distortion coefficients were estimated by self-calibrating bundle adjustment [41] with constraints imposed on the ground coordinates, using the calibration multi-camera software [42]. The mathematical model was based on the collinearity equations with additional parameters of the Conrady-Brown model [43]. In this self-calibration technique, a minimum set of seven constraints is fixed to define an object reference system, which avoids error propagation because GCPs are not required. The estimated value for the focal length was 24.27 mm, and the estimated standard deviation of the focal length and principal point coordinates were less than one pixel. The *a posteriori* variance factor (also known as estimated reference variance, which is calculated from the residuals and the redundancy, as shown by Mikhail et al. [38]) was 0.42 (with an *a priori* variance factor = 1).

### 3.2. Cylinder

As shown in Figure 4a, a PVC cylinder (25.0 cm height and 7.5 cm diameter) was used for the technical application and validation because a cylindrical surface is the object to be modelled in the proposed approach. A layer that featured a speckle pattern was used to wrap the cylinder. In addition, a strip formed by a sequence of black and white squares was used for validation purposes, which enabled the sub-pixel extraction of image coordinates from the strip that covers the entire circumference.

The cylinder was positioned on the panel with Aruco targets to enable the 3D point generation and to allow an accuracy assessment. Two images were acquired from the same planimetric position but at different heights. Figure 4b,c shows the two images used for the experiments. Both images were collected at a distance of 1.10 m from the object, having a ground sample distance (GSD) or pixel size in object space units of approximately 0.2 mm and with a base-to-distance ratio (B/D) of 0.21. The corners of the Aruco targets were automatically extracted with sub-pixel precision and were used as tie points in the self-calibrating bundle adjustment to estimate the intrinsic and extrinsic camera parameters, and 3D coordinates of each corner. Extrinsic or relative orientation can be estimated without targets using only existing texture of the environment [44], provided that scale is determined and a reference system is defined.

### 3.3. Applying Geometric Transformation

Figure 5a,b shows the effect of the viewpoint change in the central strip. The same strip appears as a curved strip in Figure 5a and as a straight strip in Figure 5b. In this case, to define the seven parameters of the geometric transformation, the corners (represented by small circles) were automatically extracted using Förstner corner detector [45].

The technique performs a geometric transformation from the image patch shown in Figure 5a to the patch shown in Figure 5b, which was considered to be a reference template. The extracted corners were used as observations in a least squares adjustment to determine the seven parameters of Equation (1). Figure 5c shows the re-sampled image patch after applying the initial geometric transformation. As can be observed, only a first approximation is obtained, as distortion effects are still visible mainly along the sides. However, this approximation of the position and shape is sufficient as an initial estimate to enable the refinement by ALSM. Figure 5d displays the result achieved with iterative adjustments by ALSM after six iterations. By comparing this result with Figure 5b, it is possible to note the similarity between the shapes.

Table 2 presents the values of the seven parameters of the geometric transformation function estimated by least squares adjustment, which resulted in an *a posteriori* variance factor of 2.66. As can be seen, *a*_1_ and *b*_1_ are the number of pixels for displacement between the image patches. The values *a*_2_, *a*_3_, *b*_2_ and *b*_3_ are the parameters that fit the geometric shape, and *b*_4_ models the concavity of the curve in Figure 5a.

As Equation (1) is composed of a set of independent parameters, the parameters can be estimated separately for *x* and *y*. The parameters *a_i_* in equation *x*′ follow the behaviour of a conventional affine transformation, and a physical interpretation (translation, rotation, scale) can be derived. In contrast, equation y′ forms a second-order polynomial that does not allow the same physical understanding because *b*_2_ and *b*_4_ are correlated. Although the parameters (*b*_i_) are independent, the final effect is produced by the combination of the three terms of the polynomial. Figure 6a–c shows graphs that represent the effects in y for each term separately as a function of the image columns, and Figure 6d,e shows the combined effects. The points used in these graphs are the corners shown in Figure 5. Because the parameter *b*_1_ represents a translation (−3698.0498 pixels), it was not graphically depicted.

As seen in Figure 6a–c, the term b_2_x produces a large vertical variation according to the column. This variation is counterbalanced by the term *b_4_x^2^* and shaped to the reference coordinates by the term b_3_y. Figure 6d indicates a parabola generated by combining both dependent terms of *x*, and Figure 6e shows the approximate fit when the term b_3_y is added to the equation y′. Figure 6f presents the differences in the columns and rows after the geometric transformation, which reduced the differences between the image patches to less than three pixels in column and less than five pixels in row. These differences were greater for the corners on the sides of the image patches, as shown in Figure 5c.

### 3.4. Continuous Matching for 3D Reconstruction

The previous step performs a pixel-by-pixel adjustment between the image patches. Thus, since each pixel has its corresponding pair defined, a mapping can be made between the image patches. Any pixel from the cylindrical surface of the first image can be mapped to its corresponding point in the second image with sub-pixel precision. The continuous 3D reconstruction is made using photogrammetric intersection. In this procedure, calibrated intrinsic parameters and extrinsic parameters of the two images are required. All pixels within the area of interest have their 3D coordinates determined using the intersection of rays by the collinearity equations. Figure 7a,b shows the 3D reconstruction of the strip from two views. Because the transformation function is defined for the entire cylinder, any part of the surface can be reconstructed continuously. Figure 7c,d displays the cylindrical reconstruction, including the textured parts of the cylindrical surface.

### 3.5. Textured Area

The first experiment used synthetic square corners that are not a natural feature. To analyse the behaviour of the proposed technique with natural objects, another experiment was conducted with only textured parts of the cylinder, which are similar to speckle patterns.

Two regions of the cylinder with different sizes (a part above the strip and another part below the strip) were used for the 3D reconstruction. The two regions or image patches were called Patch I (above) and Patch II (below) for identification in the experiments. In this case, the scale-invariant feature technique (SIFT) as proposed by [46] was applied to automatically extract the keypoints for image matching, as shown in Figure 8a,b, to estimate the initial transformation parameters. Several distributed points were used to define the preliminary seven parameters of the geometric transformation function. Table 3 presents the parameters that were estimated by least squares adjustment using the set of points of each region. Next, the refinement by ALSM was conducted, and the resulting 3D reconstructions for each region are shown in Figure 8c,d. As can be noted, the entire regions were continually reconstructed.

By analysing the parameters that result from the transformations of the two patches, the parameters *a*_1_ and *b*_1_ indicate the translations between each pair of patches. The parameter related to the concavity of the cylindrical curve is denoted by *b*_4_ and was the same in both patches, as was expected. The parameter *b*_4_ = −0.0007 also resulted in a value close to the value of *b*_4_ presented in Table 2, when a strip with synthetic squares was used, which covered the cylindrical circumference. Similarly, the parameters *a*_2_, *a*_3_, *b*_2_ and *b*_3_ also demonstrated behaviours similar to those presented in Table 2.

## 4. Analysis of the Results

The analysis of the results was based on the comparison between the diameter values estimated by both photogrammetry and direct measurement with a calliper (σ = 0.05 mm). The accuracy assessment provided by the proposed technique was determined by the difference between these two measurements. To calculate the cylinder diameter from the 3D reconstruction, a circle was adjusted to the XY coordinates. The estimated points were used as observations in the circle fitting by the least squares method.

A circle in 2D can be described by three independent parameters in which (*X_c_*, *Y_c_*) are the coordinates of the circle centre and *r* represents its radius. A point (*X_i_*, *Y_i_*) on the circle satisfies Equation (7) [38]:(7)$${\left(X_{i}-X_{c}\right)}^{2}+{\left(Y_{i}-Y_{c}\right)}^{2}=r^{2}$$

The circle fitting was solved by the iterative least squares method, adopting the centroid of the planimetric coordinates and its mean radius as initial approximations to *X_i_*, *Y_i_* and *r*. The diameter assessment with circle fitting was first performed using the strip with black and white squares over the cylinder. A comparison was also performed with the results produced by three 3D point determination techniques (see Figure 9). In all cases, the corners were automatically extracted from the two images and used as follows:*Photogrammetric intersection*: using only the 3D corner coordinates determined by the intersection of rays with previously estimated extrinsic parameters.*Bundle block adjustment (BBA)*: using only the 3D corner coordinates determined by bundle adjustment. In this case, the image coordinates of the corners were defined as tie points in the set of 52 images used in the camera calibration procedure.*ALSM + intersection*: using all 3D points (only planimetric coordinates) of the central line of the reconstructed strip, which were generated by a photogrammetric intersection with the proposed technique.

Table 4 presents the standard deviation (σ) of the three parameters that result from the circle fitting. In the three techniques, the values were less than 1 mm, notably when the combination of ALSM + intersection was used, which resulted in the smallest standard deviation values. The dense point sequence produced the smallest dispersions. The accuracy was also calculated by the difference between the estimated and measured diameters. The photogrammetric intersection and BBA techniques indicated small differences of 0.20 mm and 0.27 mm, respectively, whereas ALSM + intersection obtained the most accurate result (0.13 mm), achieving approximately 1/10 mm for a camera-to-the-object distance of 1.10 m and a base/depth ratio of 0.21.

The graph in Figure 10 displays the residuals that result in the XY coordinates after the circle fitting using 335 points of an arc segment generated by the ALSM + intersection techniques. The largest residuals were produced in the cylinder borders (>1.5 mm), where the deformation is expected to be larger. In the central part, the residuals were less than 0.9 mm. Furthermore, the residuals in both coordinates had an alternating behaviour around zero.

The reconstruction accuracy was also assessed for the textured areas presented in Section 3.5, where two image patches (above and below the strip) were considered in the experiments. For this case, two assessments on the measurement errors were made: a first using only the keypoints automatically extracted with the SIFT technique and the other by using the continuous reconstruction technique. In the first assessment, the 3D coordinates of the keypoints were estimated only with a photogrammetric intersection, and then a circle was fitted. In the second, the ALSM + intersection was used, and the XY coordinates of the 3Dpoints from each image patch were used to fit a circle, as shown in Figure 11a,b.

Table 5 presents the standard deviations and errors that result from the circle fitting in the two textured areas extracted from the cylinder, which were called Part I and Part II. The errors were computed by comparing the estimated diameters with the value that was measured directly with a calliper. When only the SIFT keypoints were used, the diameter error was 0.003 mm in Part I and 0.0079 mm in Part II. The largest standard deviations indicated in the centre circle determination a dispersion of approximately 0.37 mm in X and 2 mm in Y, with 1.74 mm for the diameter. However, more accurate values were obtained with the ALSM technique, which generates errors of 0.0001 mm and 0.0033 m for Parts I and II, respectively. In addition, the standard deviations were also smaller and in both Parts I and II were below 0.20 mm. The results with the textured area were more accurate than those achieved with the checkered strip because in those patches, the gradients are concentrated in a few directions, and the internal areas are homogenous and did not contribute to improving the solution with the modified LSM. It should be noted that the errors that were assessed by comparing the indirect estimates from the images with the direct measure were smaller than the nominal precision of the calliper, which is 0.05 mm. Thus, the error values presented in Table 5 can be considered to be optimistic, but it can be concluded that the proposed technique can deliver accuracies that are higher than 0.05 mm.

Additional experiments were performed with two metallic cylinders to assess the repeatability of the technique and its accuracy. Both cylinders were firstly painted with matte coating to avoid reflections with black ink randomly sprayed generating a speckle pattern and they were placed on the calibration panel for image acquisition, as shown in Figure 12. Next, the 3D reconstruction technique was applied as presented with the textured areas. Two calibrated electronic micrometres (Coolant Proof IP65-MX, Mitutoyo, Chicago, IL, USA, both with resolution of 0.001 mm and accuracy of 0.002 mm) were used to measure the diameters at two perpendicular positions.

Table 6 presents the direct measures of the cylinders used as reference and the results after circle fitting. As can be seen, the errors were less than 0.07 mm in the two cylinders and the standard deviations of the estimated circle parameters were less than 0.21 mm. In all cases, the standard deviations in X were smaller than in Y. This effect was caused by the camera displacement that was always performed in the *Y*-axis. Consequently, the largest deformations also occurred in the Y-direction.

For an overall assessment, the experiments with the strip and textured areas showed that the image patches were continuously reconstructed in the object space, and the accuracy achieved a sub-millimetre level, which was also confirmed with the metallic cylinders. With regard to the technical feasibility of the approach, since the extrinsic parameters of the images are known, homologous regions of the cylindrical surface can be easily located using the points extracted from the cylinder. The image coordinates of these points can be used in collinearity equations to determine the conjugated regions from which the image patches were extracted. Typically, a 3D reconstruction using photogrammetric procedures requires a large set of images. However, in this approach, only two images were needed to produce a continuous 3D reconstruction. Even regions with homogeneous textures were reconstructed, as was accomplished with the checkered strip. Some guidelines based on the produced experiments can be recommended to guarantee better technical performance:For determining the geometric transformation function, a narrow strip that covers the cylindrical circumference is sufficient.Points extracted from the surface for image matching should be distributed along the cylindrical width for modelling its curvature.Points extracted for image matching should not be aligned to better estimate the geometric transformation parameters.

In addition to these recommendations, it was observed in the experiments that there is no need for a high point density to determine the geometric transformation parameters. This finding was verified by the results produced with the checkered strip (with 20 corners) and the texture area below the strip (with 11 keypoints), which produced an accuracy close to 1/10 mm. The preliminary geometric transformation, which is a fundamental step, performs only the first approximation between the corresponding image patches at the pixel level. The ALSM adjustment is the key step that determines the match points with high accuracy.

## 5. Conclusions

This paper introduced a technique for continuous 3D reconstruction of cylindrical surfaces. The concept was based on image acquisition with a base path parallel to the axis of the cylindrical object. The concept was that changes in the camera viewpoint in a single direction would generate different cylindrical perspective effects similar to a conic section and that this effect could be modelled by a single additional parameter in a geometric transformation. Then, the different effects could be corrected by a geometric transformation followed by ALSM to produce a 3D object reconstruction using a continuous matching with only two images.

The experiments were conducted with a textured cylinder that included a checkered strip for validation purposes. Synthetic checkered corners were automatically extracted and used in the experiments to define the parameters of a geometric transformation function and also to assess the results. Keypoints that were automatically extracted from textured areas were also used to verify the technical feasibility when well-defined corners were not available.

Since the object of study was a cylinder, the accuracy assessment of the presented technique was based on the error discrepancy in determining the cylindrical diameter using circle fitting by the least squares algorithm. The diameter estimated with the proposed technique was compared to the value directly measured in the cylinder for accuracy assessment. The error obtained with the checkered strip resulted in 0.13 mm, and when the textured areas were used, the error was less than 0.012 mm. This difference between the results is due to the background of the areas. In the checkered strip, only the gradient points over the edges contribute to the solution, and several of them are correlated. On the other hand, in the textured areas, there is much variability in all directions, which provides a better refinement in the ALSM. In general, the results demonstrated that the technique achieved an accuracy of approximately 1/10 mm (for the scale and B/D ratio used in this study) considering the cylindrical circumference. Additionally, the objective of performing a continuous 3D reconstruction was achieved.

The proposed technique was tailored for a cylindrical surface, which is a common shape that is difficult to model by conventional photogrammetric techniques. However, several benefits can be envisioned. The technique is entirely developed in automatic mode and achieves sub-millimetre accuracy, if accurate extrinsic parameters are provided. The image acquisition technique is simple to handle due to the use of a single camera. The 3D reconstruction is made with a minimum of two images, and textured areas can be reconstructed. The technique can be used in industrial applications in which accurate measurements are required; in forests where trunks are commonly measured and reconstructed; or even poles in outdoor images. It is important to emphasize that the main focus of this paper was to present the modelling technique with the methodological steps and the resulting accuracies in the cases studied. The accuracy assessment that the technique can achieve in general applications has not been addressed and should be assessed in future works.

## Figures and Tables

**Figure 1 sensors-18-04183-f001:**
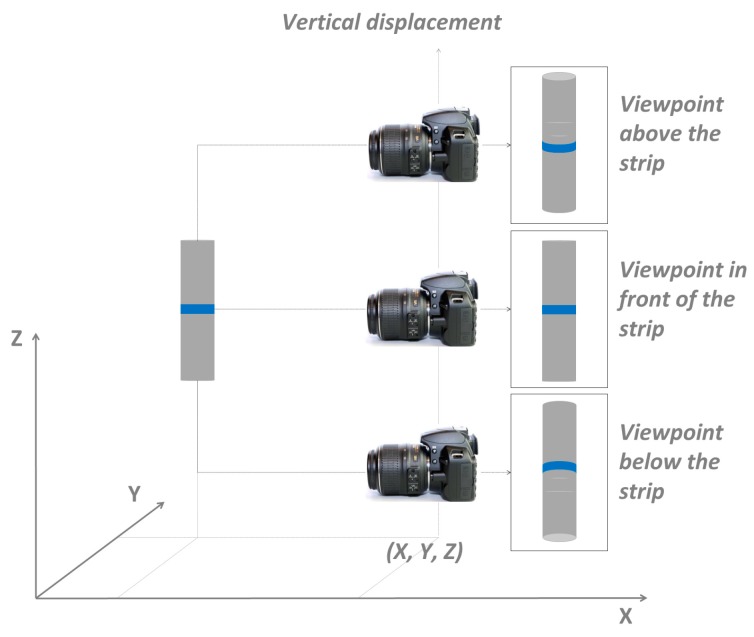
Image acquisition using camera displacement in a line path parallel to the cylindrical axis.

**Figure 2 sensors-18-04183-f002:**
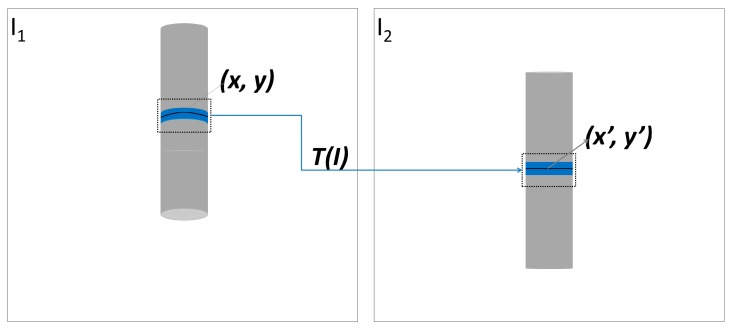
A geometric transformation function T is defined to map and approximate the corresponding points between image patches.

**Figure 3 sensors-18-04183-f003:**
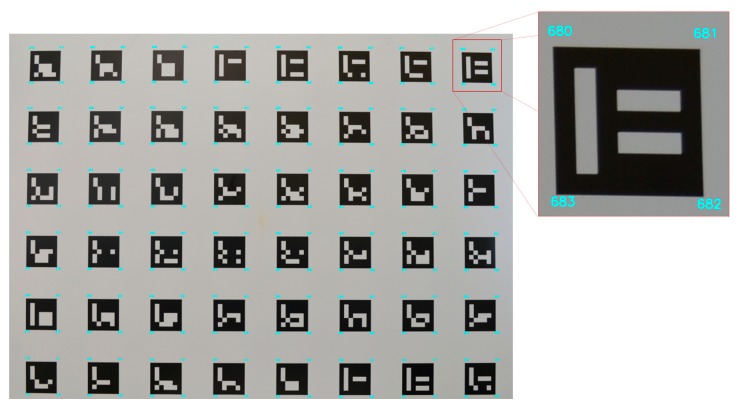
Panel composed of Aruco targets used for camera calibration, showing the corners that were automatically extracted.

**Figure 4 sensors-18-04183-f004:**
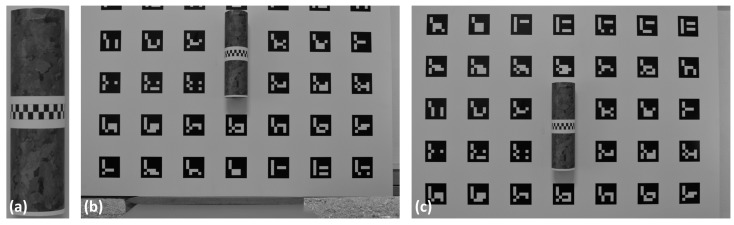
(**a**) Cylinder with a speckle pattern and with a strip covering its circumference. Two images taken from two different heights are shown in (**b**,**c**), considering that the cylinder was fixed on the calibration panel.

**Figure 5 sensors-18-04183-f005:**
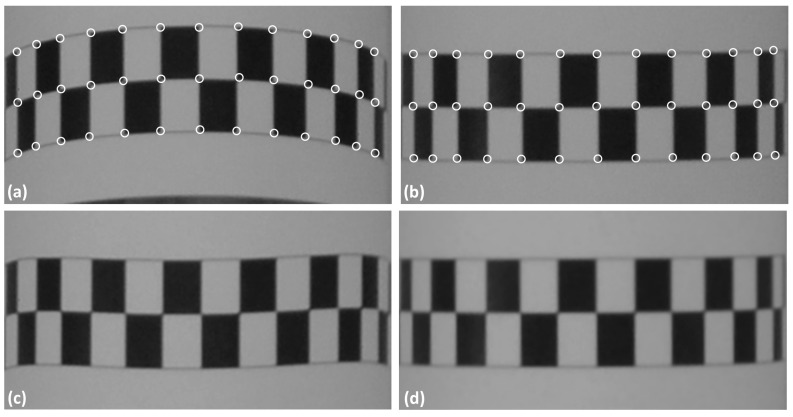
(**a**,**b**) are the resulting effects due to the camera viewpoints shown in Figure 4b,c, and the corners (white circles) automatically extracted from the corresponding image patches are also shown. (**c**) Result of the geometric transformation to fit (**a**,**b**). (**d**) Final result after applying ALSM.

**Figure 6 sensors-18-04183-f006:**
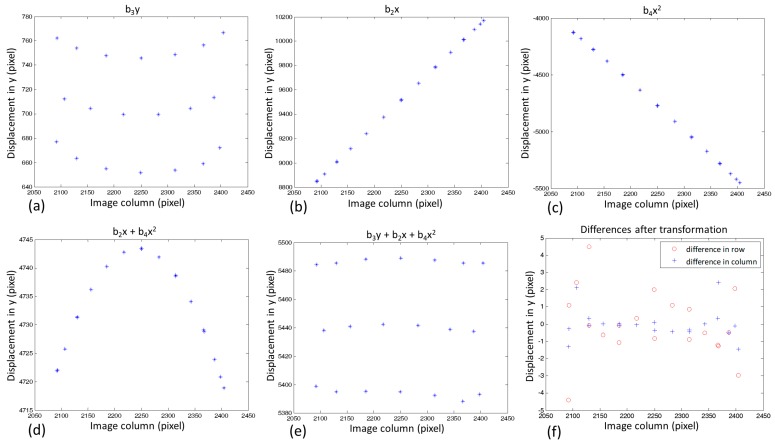
Behaviour of the polynomial terms in equation y′ (individually and combined) as a function of the image columns in (**a**–**e**), and the differences in the coordinates after geometric transformation in (**f**).

**Figure 7 sensors-18-04183-f007:**
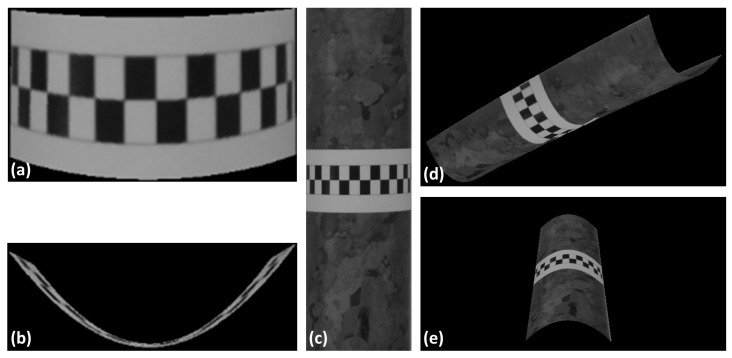
Continuous 3D reconstruction: (**a**,**b**) are two views of the reconstructed strip. (**c**–**e**) contains three views that show the cylinder reconstructed with its texture.

**Figure 8 sensors-18-04183-f008:**
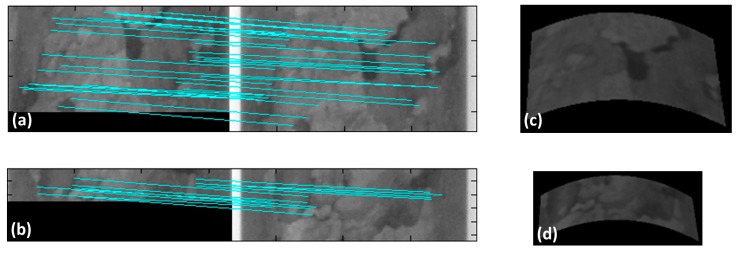
SIFT matching of the image patches using texture: (**a**) image clipping above the strip and (**b**) image clipping below the strip. Their respective 3D reconstructions are presented in (**c**,**d**).

**Figure 9 sensors-18-04183-f009:**
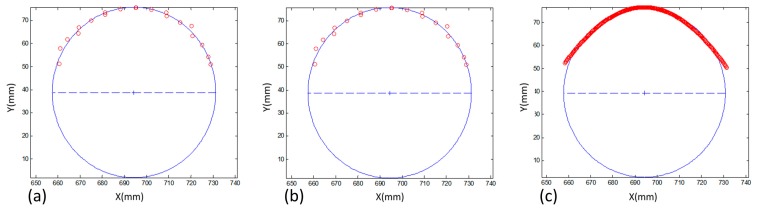
Circle fitting in the planimetric coordinates estimated with three different techniques: (**a**) photogrammetric intersection; (**b**) BBA and (**c**) ALSM + intersection.

**Figure 10 sensors-18-04183-f010:**
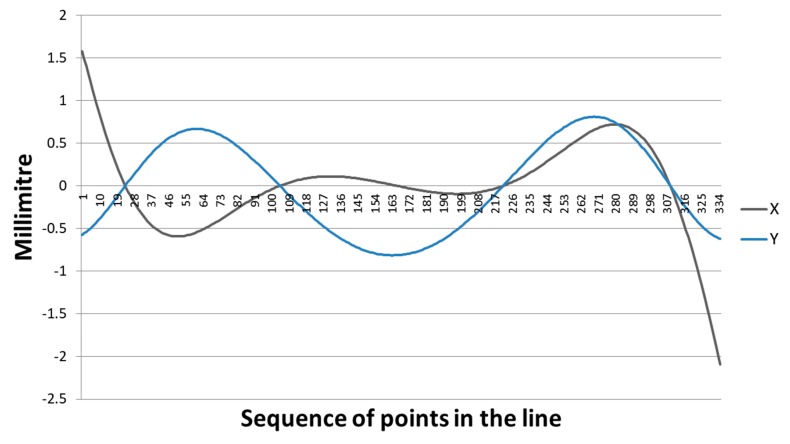
Residuals in the XY coordinates resulting from the circle fitting by least squares adjustment.

**Figure 11 sensors-18-04183-f011:**
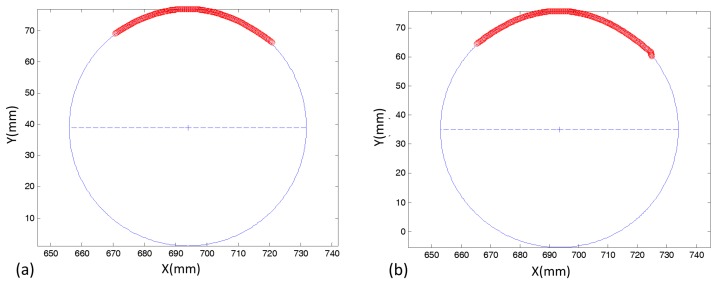
Circle fitting in the textured areas: (**a**) above the strip and (**b**) below the strip.

**Figure 12 sensors-18-04183-f012:**
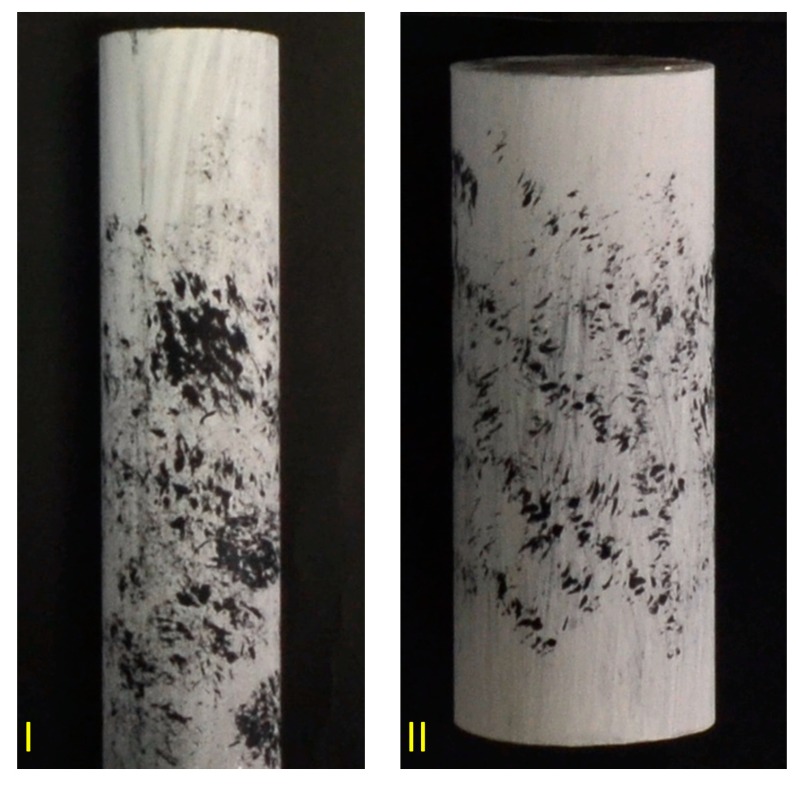
Metallic cylinders used for an additional set of experiments. (**I**) Cylinder with length 170 mm and diameter 31 mm; (**II**) Cylinder with length 126 mm and diameter 50 mm.

**Table 1 sensors-18-04183-t001:** Camera details.

Feature	Specification
Digital camera model	Nikon D3100
Sensor size	CMOS APS-C (23.1 mm × 15.4 mm)
Image dimensions	4608 × 3072 pixels
Pixel size	5 μm
Nominal focal length	24 mm (AF-S DX Nikkor)

**Table 2 sensors-18-04183-t002:** Parameters of the geometric transformation estimated by least squares adjustment.

*a*_1_ (pixel)	*a* _2_	*a* _3_	*b*_1_ (pixel)	*b* _2_	*b* _3_	*b* _4_
−72.0392	1.0112	−0.0057	−3698.0498	4.2292	1.0228	−0.0009

**Table 3 sensors-18-04183-t003:** Parameters of the geometric transformation estimated by least squares adjustment in two image patches.

Part	*a* _1_	*a* _2_	*a* _3_	*b* _1_	*b* _2_	*b* _3_	*b* _4_
Patch I	−78.9782	1.0018	0.0510	−2582.1701	3.2421	0.9983	−0.0007
Patch II	−95.1206	1.0085	0.0297	−2522.9268	3.2021	1.0013	−0.0007

**Table 4 sensors-18-04183-t004:** Estimated parameters with the circle fitting using planimetric coordinates provided by the three techniques.

Photogrammetric Technique	Standard Deviation (mm)			Difference between Diameters (mm)
	σ_Xc_	σ_Yc_	σ_diameter_	
Intersection	0.3016	0.8686	0.6773	0.20
BBA	0.3019	0.8697	0.6782	0.27
ALSM + Intersection	0.0702	0.1984	0.1623	0.13

**Table 5 sensors-18-04183-t005:** Circle parameters estimated for the two textured areas.

Textured Area	Standard Deviation (mm)			Diameter Error (mm)
	σ_Xc_	σ_Yc_	σ_diameter_	
Part I—only SIFT keypoints	0.3286	1.4588	1.2308	0.0030
Part II—only SIFT keypoints	0.3708	2.0669	1.7389	0.0079
Part I—ALSM	0.0702	0.1983	0.1622	0.0001
Part II—ALSM	0.0302	0.1038	0.0916	0.0033

**Table 6 sensors-18-04183-t006:** Measures of the two metallic cylinders and results of the circle fitting.

Cylinder	Length (mm)	Measured Diameter (mm)	Standard Deviation (mm)			Diameter Error (mm)
			σ_Xc_	σ_Yc_	σ_diameter_	
I	170	31.702	0.028	0.207	0.207	0.066
II	126	50.622	0.026	0.083	0.074	0.050

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
