# Peer review of "Reconstruction of Cylindrical Surfaces Using Digital Image Correlation"

_sensors, 2018, doi:10.3390/s18124183_

Round 1
Reviewer 1 Report
This paper proposes a new method for dense accurate photogrammetric reconstruction of cylindrical surfaces.
The key assumption of this work is that of already knowing that the considered object has a cylindrical surface.
Given this assumption, the authors properly models assumptions made on camera system and object geometry.
The paper is interesting and well written. Actually I only have some minor comments:
- page 7, line 229 and page 8, line 272: I think it is probably better if authors specificy the used definition for a posteriori variance factor.
-page 13, Table 5: diameter error for Part II - ALSM is surprisingly much larger than in Part I (0.0033mm vs 0.0001). Despite having a large error there is somehow expected (sift keypoint case also had a larger error in part II than I), having such a big difference between the cases is relatively strange. Please, provide some explanation about such big difference if possible.
Author Response
Response to Reviewer 1 Comments
This paper proposes a new method for dense accurate photogrammetric reconstruction of cylindrical surfaces. The key assumption of this work is that of already knowing that the considered object has a cylindrical surface. Given this assumption, the authors properly models assumptions made on camera system and object geometry. The paper is interesting and well written. Actually I only have some minor comments:
- pag 7, line 229 and pag 8, line 272: I think it is probably better if authors specificy the used definition for a posteriori variance factor.
Response:
A sentence was included with a reference:
“… (also known as estimated reference variance, which is calculated from the residuals and the redundancy as shown by Mikhail et al. [38]).”
-pag 13, Table 5: diameter error for Part II - ALSM is surprisingly much larger than in Part I (0.0033mm vs 0.0001). Despite having a large error there is somehow expected (sift keypoint case also had a larger error in part II than I), having such a big difference between the cases is relatively strange.
Please, provide some explanation about such big difference if possible.
Response:
The direct measurements of the first experiment, used as references, were done independently by a third partner professional with an mechanic calliper, which has a precision of 0.05 mm, as it was described in the paper.
Thus, it is expected that the standard deviation of the errors are higher than this value because of the error propagation of the two techniques. We just compared our indirectly determined value with the value measured by the third partner professional. This error is smaller than the error in the measurement of the reference diameter, so we can conclude that it was an optimistic random coincidence, as it was mentioned in the manuscript in the paragraph before Table 5.
Reviewer 2 Report
The work is well presented, well written, the method is evaluated, but very limited in the application, which leaves doubts about its value for publication. If we think of applying this to pipes, for instance, it will probably fail, because the problem is to determine the exterior orientation and the relative orientation, not to reconstruct the 3D surface, which is already solved. You may argue that you used a new equation, with an extra parameter, but in practice, you will have to determine those parameters, together with the orientation parameters, for every pair of photos of your coverage. The use of the targets for determining the orientations is not feasible in the praxis. Pipes are long features that you can not photograph in the lab. In addition, Photogrammetric intersection only needs two images, so the fact that you recovered the surface only with two images is not new, actually it is more than a hundred years old.
I would suggest the authors to rethink the objective of this experiment. Perhaps complement it with the reconstruction of other curved surfaces. Present a solution for determining also the orientation of several pairs. Nevertheless, the work is fine, written in perfect english, almost fully understandable. Only the approach seems somewhat aimless because of the weak premises.

Author Response
Response to Reviewer 2 Comments
The work is well presented, well written, the method is evaluated, but very limited in the application, which leaves doubts about its value for publication. If we think of applying this to pipes, for instance, it will probably fail, because the problem is to determine the exterior orientation and the relative orientation, not to reconstruct the 3D surface, which is already solved. You may argue that you used a new equation, with an extra parameter, but in practice, you will have to determine those parameters, together with the orientation parameters, for every pair of photos of your coverage. The use of the targets for determining the orientations is not feasible in the praxis. Pipes are long features that you can not photograph in the lab. In addition, Photogrammetric intersection only needs two images, so the fact that you recovered the surface only with two images is not new, actually it is more than a hundred years old.
Response: Many thanks for your comments. We agree that the proposed technique is very specialized to a specific object, but this was the purpose of the development. We expect that this algorithm should be part of an interactive measurement system in which further algorithms for different objects would be integrated. Extrinsic or relative orientation parameters can be estimated without targets using only existing texture of the environment (Stamatopoulos et al., (2012)), provided that scale is determined and a reference system is defined (seven constraints). Scale can be determined either by fixing the base length or by measuring a distance in the object space. In the experiments, several coded targets were used to ensure high accuracy and to enable comparison with other techniques. In the production environment, orientation could be done without coded targets.
Regarding the number of images, we fully agree with you. It was exactly what we showed in our experiments: very accurate measurements can be done with a single stereo pair instead of using dozen of images.
A sentence and a reference were included in Section 3.2:
Extrinsic or relative orientation can be estimated without targets using only existing texture of the environment [44], provided that scale is determined and a reference system is defined.
I would suggest the authors to rethink the objective of this experiment. Perhaps complement it with the reconstruction of other curved surfaces. Present a solution for determining also the orientation of several pairs. Nevertheless, the work is fine, written in perfect english, almost fully understandable. Only the approach seems somewhat aimless because of the weak premises.
Response: The reconstruction of other curved surfaces is feasible with a similar approach, but it requires assessments with more experiments that are beyond the scope of this paper. This suggestion is left as a recommendation for future works.
The last sentence of the conclusion was rewritten as:
“The accuracy assessment that the technique can achieve in general applications and for other curved surfaces have not been addressed and should be assessed in future works.”
Page: 1
squares
Response: the word was corrected.
Page: 4
The reference list goes only until 37. Every reference hereafter is not listed in your References.
Response: Sorry, there was a problem with citation management software in updating the list of references. Now, the list is correct.
Page: 5
This means there must be a texture on the cylinder
Response: Yes, some kind of texture is required to extract points that describe the surface curvature to enable the parameter estimation. Such texture could also be produced using structured light projection, for example. This speckle painting technique is quite common in machine vision (see, for instance, Pan et al. 2013).
The following sentence was inserted in the paragraph:
“Some kind of texture is required to extract points that describe the surface curvature, which could be produced either by a speckle painting [33] or by structured light projection.”
Page: 12
This approach is not clear. A line was extracted to fit a circle?
Response: The line was only used to display the graphs in Figure 11.
The sentence was rewritten:
In the second, the ALSM + intersection was used, and the XY coordinates of the 3D points from each image patch were used to fit a circle, as shown in Figures 11 (a, b).
SIFT
Response: Ok, it was rewritten in capital letters.
Page: 14
well... the minimum to 3D photogrammetric reconstruction is 2 images, as long as you know the interior and the exterior orientation parameters. Recent algorithms use much more images in order to calculate those parameters, not for the 3D reconstruction. The amount of images is also neccessary in order to discard false matchings and avoid occultations.
Response:
Current approaches have used a large number of images to reconstruct objects. In this study, the purpose was to develop a mathematical function to model the perspective effect on the cylindrical surface from the change in the camera viewpoint. Then, a single stereo pair provides very accurate results. Nevertheless, a similar approach can be used for several images, probably resulting in higher accuracy (this option has not been tested yet).
This depends from the scale. You were certainly very close to the object
Response: Yes, the accuracy is scale dependent and this is expected in photogrammetry. Final precision depends on scale and base/distance (B/D) ratio, as it was mentioned. For the same B/D, the accuracy would be proportional to scale. The experiments presented in this work simulated distances compatible with industrial environments.
Page: 15
scale dependent
Response:
A sentence was included:
“(for the scale and B/D ratio used in this study)”
Round 2
Reviewer 2 Report
I maintain my doubts about applicability because one can not only see one part of the problem.
But, as I said, the quality of the paper is good and I accept partially the answers of the authors.